# Exploring the Interplay of Handgrip Neuromuscular, Morphological, and Psychological Characteristics in Tactical Athletes and General Population: Gender- and Occupation-Based Specific Patterns

**DOI:** 10.3390/jfmk10010022

**Published:** 2025-01-04

**Authors:** Miloš M. Milošević, Nenad Koropanovski, Marko Vuković, Branislav Božović, Filip Kukić, Miloš R. Mudrić, Andreas Stamatis, Milivoj Dopsaj

**Affiliations:** 1Faculty of Physical Education and Sports Management, Singidunum University, 11000 Belgrade, Serbia; 2Department of Criminalistics, University of Criminal Investigation and Police Studies, 11000 Belgrade, Serbia; korpan82@gmail.com; 3Academy for National Security, 11000 Belgrade, Serbia; cipiripi.judo@gmail.com; 4Faculty of Sport and Physical Education, University in Belgrade, 11000 Belgrade, Serbia; branislav.bozovic.91@gmail.com (B.B.); milos.mudric@fsfv.bg.ac.rs (M.R.M.); milivoj.dopsaj@gmail.com (M.D.); 5Faculty of Physical Education and Sports, University of Banja Luka, 78000 Banja Luka, Bosnia and Herzegovina; filip.kukic@gmail.com; 6Health & Sport Sciences, University of Louisville, Louisville, KY 40292, USA; coach_stam@rocketmail.com; 7Sports Medicine, University of Louisville, Louisville, KY 40292, USA

**Keywords:** handgrip strength, maximal force, rate of force development, body mass index, mental toughens, dark triad

## Abstract

Background/Objectives: The correlation of handgrip strength (HGS) and morphological characteristics with Big Five personality traits is well documented. However, it is unclear whether these relationships also exist in highly trained and specialized populations, such as tactical athletes, and whether there are specific differences compared to the general population. This study aimed to explore the interplay of handgrip neuromuscular, morphological, and psychological characteristics in tactical athletes and the general population of both genders. Methods: The research was conducted on a sample of 205 participants. A standardized method, procedure, and equipment (Sports Medical solutions) were used to measure the isometric neuromuscular characteristics of the handgrip. Basic morphological characteristics of body height, body mass, and body mass index were measured with a portable stadiometer and the InBody 720 device. Psychological characteristics were assessed with the Mental Toughness Index and Dark Triad Dirty Dozen questionnaires. Results: Numerous significant correlations were obtained, as well as differences between tactical athletes and the general population of both genders. The most prominent correlations were between the excitation index with Psychopathy and the Dark Triad (ρ = −0.41, −0.39) in female tactical athletes, as well as Neuroticism with body height, maximal force, and the maximum rate of force development in the male general population (ρ = 0.49, 0.43, 0.41). The obtained results also revealed gender and occupational specific patterns of researched relationships. Conclusions: Although the results of this study indicated the possibility of the existence of correlations between handgrip neuromuscular, morphological, and psychological characteristics in tactical athletes of both genders, nevertheless, at the moment, there is not enough solid evidence for that. That is why new research is needed. An analysis of muscle contractile and time parameters as neuromuscular indicators in the HGS task proved to be a possible promising method, which brought numerous new insights about the researched relationships. For practical application in the field, we propose including Mental Toughness and the Dark Triad traits in the selection process for future police officers and national security personnel based on the obtained results.

## 1. Introduction

The *handgrip strength* (HGS) test is an effective marker of overall body strength and health [1] used in the periodical testing and selection of athletes [2]. It is a highly body lateralized and gender dimorphic measure [3], a reflection of muscular force and neural activity (i.e., neuromuscular characteristics) [4]. The correlation of HGS with psychological characteristics such as the *Big Five* (B5) personality traits is well-documented [5,6,7,8,9]. A gender-specific pattern in these relationships has also been identified [7]. Observed associations could be explained by evolutionary mechanisms of adaptation and natural selection [3]. However, all prior research has been conducted on the general population. What is not known is whether these relationships exist and whether they are specific in strictly selected populations that are characterized by long-term and strenuous physical exercise. The question arises: Since physical exercise affects neuromuscular characteristics, would the relationships of HGS and psychological characteristics still be present among tactical athletes such as police and national security officers? Furthermore, studies so far have not analyzed the discussed associations according to the force-time curve produced in the HGS task. This means that, apart from *maximal force* (F_max_), the associations between psychological characteristics and *the maximum rate of force development* (RFD_max_), *the time required to achieve F_max_* (tF_max_), and *the time required to achieve RFD_max_* (tRFD_max_)—all of which can be derived from analyzing the force-time curves produced during the HGS task—have not been explored in the previously mentioned studies. Moreover, indexes of dimorphism and excitation, which are also important neuromuscular characteristics [2] and can be derived from basic measures of the HGS task, have also never been studied in association with neuromuscular and psychological characteristics. It should be emphasized that described method of the force-time curve analyzing in the HGS task is reliable and widely used in the testing of athletes [10]. Also, it is well documented that various physiological mechanisms are responsible for the manifestation of different neuromuscular characteristics [11,12]. This gives the rationale for conducting new expanded research.

Basic morphological characteristics of *body height* (BH), *body weight* (BW), and *body mass index* (BMI) are important in understanding and predicting health status and professional performance [13,14,15]. They are highly associated with neuromuscular characteristics [16] and also with those measured using HGS [17,18]. Moreover, they are correlated with psychological characteristics such as the B5 personality traits [19] and can also mediate the already described relationships of HGS and psychological characteristics [7]. Although some research was conducted with military veterans [19], the relationships of morphological and psychological characteristics in various groups of tactical athletes have not been sufficiently investigated so far.

Regarding psychological characteristics, recent research suggests that personality encompasses dimensions beyond those captured by the B5 personality traits and that broadening the domain of personality assessment with *Mental Toughness* (MT) and the *Dark Triad* (DT) is necessary for the valid prediction of stress-related reactions and behaviors [20]. While MT is a psychological resource that enables individuals to achieve an optimal level of performance despite stressors of any kind [21], DT consisting of personality traits of *Narcissism* (Nrc), *Machiavellianism* (Mch), and *Psychopathy* (Psc) represents a tendency for a lack of empathy, manipulation, and the exploitation of others, an absence of morality, unemotional callousness, antisocial behavior, impulsivity, selfishness etc. [22]. Both MT and DT have a significant role in coping with stress, stress tolerance, and stress reactions [23,24]. Moreover, MT and DT carry valuable information about perceived stress [20], which is key for understanding and predicting prolonged stress reactions [25]. Keeping in mind that, due to the constant exposure to threats, tactical athletes rank among the most high-stress professions [25,26,27], the expansion of previous research on the association between morphological, neuromuscular, and psychological characteristics with MT and DT is justified.

Research into gender- and occupation-based specifics of the relationship between morphological and handgrip neuromuscular with psychological characteristics, in addition to the theoretical contribution in filling in the gaps in the description of these relationships in specific populations, also has the potential to bring new insights into the possible mechanisms underlying these relationships. Moreover, this research can have an applied value for the selection and training of tactical athletes, since all three domains of characteristics are related to health outcomes and job performance.

This study aimed to explore the interplay of handgrip neuromuscular, morphological, and psychological characteristics in tactical athletes and the general population of both genders. The first hypothesis (H1) was that the tactical athletes and general population of both genders differ according to the handgrip neuromuscular, morphological, and psychological characteristics. The second hypothesis (H2) was that handgrip neuromuscular and morphological characteristics are associated with psychological characteristics in both tactical athletes and the general population. Finally, third hypothesis (H3) was that the observed associations show gender- and occupation-based specifics.

## 2. Materials and Methods

### 2.1. The Participants

The research was conducted on a convenience sample of 205 participants (age 22.91 ± 7.41 in years., BH = 1.76 ± 0.09 in cm, BW = 73.02 ± 13.04 in kg, BMI = 23.47 ± 2.91 kg/m^2^), including 93 females (age 22.25 ± 6.25, BH = 1.69 ± 0.06, BW = 64.25 ± 9.76, BMI = 22.51 ± 3.07) and 112 males (age 23.46 ± 8.24, BH = 1.82 ± 0.06, BW = 80.30 ± 10.75, BMI = 24.27 ± 2.52), as well as 136 tactical athletes (age 20.12 ± 3.44, BH = 1.77 ± 0.08, BW = 73.41 ± 11.90, BMI = 23.33 ± 2.73) and 69 members of the general population (age 28.41 ± 9.75, BH = 1.73 ± 0.11, BW = 72.26 ± 15.11, BMI = 23.76 ± 3.25). Approximately 12 participants were left-handed and 1 was ambidexterous, while all others where right-handed. Participants were students of the University of Criminal Investigation and Police Studies in Belgrade, Academy for National Security in Belgrade, members of the general student population (predominantly from the Faculty for Special Education and Rehabilitation and the Faculty of sport and physical education, University of Belgrade), as well as members of the working population of different professional orientations. Participants from the general population were volunteers who responded to an invitation placed by the researchers on social networks. Regarding participation in sport, 25.35% members of the general population group declare themselves as non-athletes, 35.21% as recreational athletes, 5.63% as active athletes, and 33.80% as former athletes. The group had 5.87 ± 6.09 years of sport and physical activity experience. The criteria for participation were voluntary registration and the absence of any health problems before and during the study. Any history of arm injury was part of the exclusion criteria. Taking into account that this was exploratory research, such a sample enabled both the observation and description of the basic relationships of the researched variables, as well as the comparison of different populations. The total sample was divided into subsamples according to gender (female–male) and occupation (tactical athletes–general population). The descriptive parameters of the subsamples are presented in Table 1 and Table 2.

### 2.2. Procedures

All testing procedures were conducted within two sessions. During the first session, participants completed the questionnaire recording socio-demographic status, as well as the *Mental Toughness Index* (MTI) and *Dark Triad Dirty Dozen* (DTDD) questionnaires, with no time limit. The second session was performed the next day and included the BH and BM measurements, as well as the performance of the HGS test. Test was preceded by a standard warm-up routine, including 5 min of upper-body exercises and 3 min of dynamic exercises that activate the tested muscle groups, which included superficial and deep finger flexor muscles (e.g., holding a 10 kg dumbbell, 5–10 s hang on a pull-up bar, 3–5 inverted rows). During the last part of the warm-up, each participant performed two HGS trials with a gradual increase of muscle force until F_max_ and two maximally strong and fast trials. The rest between the warm-up trials was 1–2 min, while the rest between the warm-up trials and the actual testing procedures was about 5 min. The HGS tests for dominant and nondominant hands were performed in random order.

### 2.3. Morphological Characteristics

BH was measured with a portable stadiometer (Swiss instruments, Zurich, Switzerland), while BM was measured on the InBody 720 device in accordance with a standardized procedure [28]. BMI was calculated from the obtained values.

### 2.4. Handgrip Neuromuscular Characteristics

The HGS was measured using the SMS HG system and software program (Isometrics Lite, ver. 3.1.1, Isometrics SMS All4Gym, Belgrade, Serbia). This system was shown to be valid and reliable for this type of testing compared to the Jamar Handgrip Dynamometer, which is considered the gold standard for handgrip testing (dominant hand ICC = 0.98, non-dominant hand ICC = 0.97) [29]. Although the validation study showed that the results obtained by two devices cannot be compared directly, and that the instruments cannot be used interchangeably, it also showed that both devices accurately measure the same neuromuscular characteristic of the muscle groups. That is why this system is in wide use in the testing of athletes, as well as in research [2,29,30]. The SMS HG system allows adjustment to each participant’s handgrip size. The HGS device was attached to the force transducer that measured the isometric force of finger flexors. The standard tensiometric probe with the measurement precision of ±0.01 N was connected to the force reader. The force-time signal was sampled at 1000 Hz and low-pass filtered (10 Hz) using a fourth-order (zero-phase lag) Butterworth filter [31]. RFD was calculated as the maximal slope of the force-time curve (over the first derivative of the force-time curve) in regards to the force onset [31]. Prior to measurement, the device was calibrated. The onset of the contraction was defined as the point in time when the first derivative of the force-time curve exceeded the baseline by 3% of its maximal value.

Participants performed the HGS test according to the previously reported procedures [3,30]. In short, participants were in a sitting position, holding the measuring device in their hand, with the arm extended next to the body, while the other arm was resting alongside the body. They were not allowed to change positions during the test or to lean their hand or the device on their thigh or another solid object.

Since the results of muscular force increase non-linearly with body size, standard allometric partialization was performed for the obtained force and explosivity by dividing it with body mass scaled to 2/3, i.e., the given results were analyzed from the aspect of relative values (F_rel_ and RFD_rel_) [32].

For the HGS test, participants were instructed to squeeze the device as firmly and as quickly as possible. The test was performed twice with a 2 min between-trial rest. Their force output was projected on the screen, and they were verbally encouraged to obtain the best result. From the HGS test, F_rel_ [N/kg^2/3^], tF_max_ [s], RFD_rel_ [N/kg^2/3^/s], and tRFD_max_ [s] of the dominant (D) and non-dominant (ND) hands were collected while they were included in further analyzes as sums of values for the D and ND hands. From collected values the DI of F_rel_ and RFD_rel_ (dimorphic index for force and explosivity), as well as EI (excitation index), were calculated as follows.

DIF = F_rel_D/F_rel_NDDIRFD = RFD_rel_D/RFD_rel_NDEI = (F_rel_D + F_rel_ND)/(RFD_rel_D + RFD_rel_ND)

### 2.5. Psychological Characteristics

MTI and DTDD questionnaires were used to assess psychological characteristics. In order to make it easier to compare the scores on different questionnaires and subscales, they were calculated as average values of the respondents’ answers to all items concerning one questionnaire or subscale. This procedure has proven to be valid in previous studies [33].

MT was assessed by MTI [21]. The instrument consists of eight items that were answered using seven-point Likert-type assessment scales. The total score varies from a minimum of 1 (False, 100% of the time) to a maximum of 7 (True, 100% of the time). The construct validity of the MTI has been supported by studies involving participants from various cultures [34]. In previous studies, the reliability estimates for MTI scores were ≥0.86 [34].

DT traits were assessed by DTDD [35,36], which consists of 12 items that were answered using seven-point Likert-type assessment scales. The DTDD assesses an individual’s overall DT through three socially malevolent traits: Mch, Psc, and Nrc. The scores vary from a minimum of 1 (poorly present) to a maximum of 7 (extremely present). In previous studies, the reliability estimates for DTDD scores were ≥0.77 [35,36].

### 2.6. Statistical Analysis

The sample size was determined after applying a power analysis. For two-tail *t*-tests (correlation: point biserial model, two tails, with α = 0.05, power 1-β = 0.80, and effect size ρ = 0.50), the sample size should comprise at least 26 participants, while for the effect size ρ = 0.30, the total sample size should comprise at least 82 participants. For two-tail *t*-tests (means: Wilcoxon–Mann–Whitney test [two groups], with α = 0.05, power 1-β = 0.80, effect size d = 0.7, and 2 groups), the sample size should be 70 participants at least. Power analyses were performed using G-power 3.1.9.6 (Franz Faul, Univesitat Kiel, Germany).

All statistical analyses were performed using SPSS 20 (IBM Corp., Armonk, NY, USA). Statistical significance was defined at the level of 95% probability for the value of *p* < 0.05 and at the level of 99% probability for the value of *p* < 0.01. A descriptive statistical analysis was performed, including mean (M), standard deviation (SD), minimal (Min), and maximal (Max) values, and coefficient of variation (CV). The Kolmogorov–Smirnov test was used to assess the normality of distribution, while Levene’s test was used to assess the homogeneity of variance.

Keeping in mind the mixed nature of the data, the violation of normal distribution in some variables, as well as the homogeneity of variance, the basic assumptions for the usage of parametric statistics were not met, so it is necessary to apply non-parametric methods in further analyses. In order to discover the differences between tactical athletes and the general population (H1), a Mann–Whitney U test was performed. To discover the relationship between the morphological and handgrip neuromuscular characteristics with psychological characteristics (H2), as well as gender- and occupation-based specific patterns (H3), a Spearman’s correlation analysis was performed on all four subsamples. The criteria for evaluation of the effect size in Mann–Whitney U test were: 0.01 < η^2^ < 0.06—small, 0.06 < η^2^ < 0.14—medium, η^2^ > 0.14—large effect [37]. The effect size of correlation coefficients was defined as 0.20 ≤ weak ≤ 0.49, 0.50 ≤ moderate ≤ 0.79 and strong ≥ 0.80 [38,39].

## 3. Results

The descriptive statistical analysis is presented in Table 1 and Table 2. The nonparametric Kolmogorov–Smirnov test showed significant deviations from the normal distribution for the variables MT, Mch, and Psc, while Levene’s test showed a significant violation of variance homogeneity in Mch, F_rel_, and RFD_rel_.

In the female subsample, both tactical athletes and the general population demonstrated high variability according to CV in Mch, Psc, Nrc, DT, and tF_max_ (Table 1). The non-parametric Mann–Whitney U test revealed significant differences (*p* < 0.05) with medium effect size (0.06 < η^2^ < 0.14) between the female tactical athletes and female general population in MT (U = 742, z = −2.4), Mch (U = 737.5, z = −2.5), BH (U = 651, z = −3.1), and F_rel_ (U = 749.5, z = −2.4).

The results of the descriptive statistical analysis for the male subsample are presented in Table 2.

Similar to the findings in women, both subsamples exhibited high variability in Mch, Psc, Nrc, DT, and tF_max_, as indicated by the CV. In the male sample, the non-parametric Mann–Whitney U test revealed significant (*p* < 0.05) differences with a large effect size (η^2^ > 0.14) in MT (U = 595.5, z = −4.2), medium effect size (0.06 < η^2^ < 0.14) in BW (U = 762, z = −3.1), tF_max_ (U = 847, z = −2.5), and DIRFD (U = 785, z = −2.9), as well as a small effect size (0.01 < η^2^ < 0.06) in BMI (U = 859, z = −2.4) and DIF (U = 917, z = −2).

When it comes to gender differences in psychological characteristics, the non-parametric Mann–Whitney U test did not show significant differences between the male and female general populations, but at the same time, revealed significant (*p* < 0.05) differences with a small effect size (0.01 < η^2^ < 0.06) in tactical athletes in MT (U = 1655.5, z = −2.5) and Psc (U = 1649.5, z = −2.5).

The results of the correlation analysis for the female subsample are presented in Table 3.

The correlation analysis in the female subsample (Table 3) revealed a weak effect association of MT with DIF (ρ = 0.27), which approached the threshold for statistical significance (*p* = 0.05), and a statistically significant, but still weak effect association of Psc, Nrc, and DT with RFD (ρ = −0.33, −0.30, and −0.35, respectively), and DT with EI (ρ = −0.39), as well as Psc with EI (ρ = −0.41), among female tactical athletes. When it comes to the general population, a statistically significant, but weak correlation of Nrc with BH was also revealed (ρ = 0.33).

The results of the correlation analysis for the male subsample are presented in Table 4.

The results of the correlation analysis in the male subsample (Table 4) among tactical athletes revealed weak effect associations of MT with BH, tRFD_max_ (ρ = 0.22 and 0.21) and the same level of statistical significance and strength of relation of Mch with tRFD_max_, DIRFD, and EI (ρ = −0.35, 0.26, and −0.25, respectively), and DT with EI (ρ = −0.27).

When it comes to the general population, weak effect correlations of MT with F_max_, Nrc with BH, F_rel_, and RFD_rel_, as well as DT with BH were also revealed (ρ = 0.37, 0.49, 0.43, 0.41, and 0.42, respectively).

Occupation-based specific patterns of the relationship of handgrip neuromuscular and psychological characteristics among tactical athletes and the general population are illustrated in Figure 1.

Differences in the slopes of the regression lines show that while an increase in EI is not followed by a change in MT in tactical athletes (B = 2.41), among the general population (B = 0.18), it is followed with an increase in MT (Figure 1a). On the other hand, unlike in the general population (B = 0.44), in tactical athletes (B = −6.22), an increase in the excitation index is followed with a decrease in DT (Figure 1b).

Occupation-based specific patterns of the relationship of morphological and psychological characteristics among tactical athletes and the general population are illustrated in Figure 2.

In contrast to EI, an increase in BMI is correlated with an increase (B = 0.02) in MT among tactical athletes, and at the same time, with a decrease (B = −0.02) in MT in the general population (Figure 2a), while changes in BMI are correlated with similar changes (B = 0.02, 0.01) in DT in both groups (Figure 2b).

## 4. Discussion

This study aimed to explore the associations between handgrip neuromuscular, morphological, and psychological characteristics among tactical athletes and the general population. The obtained data revealed significant differences between the investigated groups across all three examined domains of characteristics, thereby supporting the first hypothesis.

To better interpret the obtained results, it is essential to first examine the descriptive parameters (Table 1 and Table 2). Comparing the morphological characteristics with the general population [13,14], it can be noticed that participants in all four subsamples have a healthy, i.e., good, nutritional status. The values of all handgrip neuromuscular variables are also above average in the general population [30]. Psychological indicators show that all four subsamples are characterized by weakly present Mch, Psc, and DT, moderate Nrc, and highly present MT. When compared to the results of previous studies [20,24], the obtained profile can be described as benevolent and stress resilient. Although our research clearly documented the difference between the subsamples of tactical athletes and the general population, in favor of the greater development of the first group in all domains, it seems that the general population group is positioned in all three domains above what was expected based on previous research. The only consistent difference with a large effect was obtained in MT. This finding could be interpreted in two ways: as a consequence of the selection of tactical athletes, but also as a consequence of the influence of training on the development of MT. Most likely, both explanations are valid and represent an opportunity for further research on this topic. Nevertheless, the results of the descriptive analysis (Table 1 and Table 2) and the Mann–Whitney U test suggest that the general population group, since they applied for the research themselves, which was conducted at the Faculty of Sports, actually has an above-average experience in sport and physical exercise. On one hand, this finding reduces the representativeness of the general population subsample and the possibility of generalization of the derived conclusions. On the other hand, since both groups exhibit similar levels of muscle strength and explosivity (Frel and RFD_rel_) as a indicators of physical fitness, the observed differences in correlations can be more clearly explained with occupational choice than with previous athletic experience. This aspect contributes to the applicability of the obtained results in the practice of selection and training of tactical athletes.

The obtained correlations (Table 3 and Table 4) are ambiguous when it comes to expectations based on prior research [5,6,7,19]. While a weak, but positive correlation of BH with MT in the male tactical athlete group, and even with Nrc in the general population groups of both genders, can be described as expected, a positive correlation with DT in the male general population comes as surprise (Table 4, ρ = 0.42). This is the first significant difference between the groups in relation to their professional orientations. When it comes to neuromuscular characteristics, the obtained relationships with MT are also largely expected. Higher Fmax in the male general population and DIF in the female tactical athletes are followed with greater MT. However, greater tRFDmax (longer time represents a measure of slower maximal muscle excitation) accompanied by an increase in MT in male tactical athletes is also surprise because shorter time actually represents superior neuromuscular functioning. The obtained correlations with DT are maybe the biggest surprise. Superior neuromuscular characteristics were associated with greater tendencies for antisocial behavior in tactical athletes of both genders. This was especially true for men in the general population group, as F_rel_ has the same direction of correlation (positive) with both MT and with the entire Dark Triad. However, it should be noted that the correlations with Msc, Psc, and DT are not significant, and any conclusions based on them should be taken with caution before more detailed investigations are conducted. This result supports the earlier conclusion that expanding the domain of personality assessment beyond the B5 traits to include MT and DT is essential for a valid understanding of predicted reactions to chronic stress [20]. This finding should be taken seriously by practitioners working on the selection of tactical athletes because some superior neuromuscular functioning is related to negative behavioral tendencies, both in men and women. The most prominent obtained correlations of EI with Psc and DT (ρ = −0.41, −0.39) in female tactical athletes, as well as Neuroticism with body height, maximal force, and maximum rate of force development in the male general population (ρ = 0.49, 0.43, 0.41), suggest that morphological and neuromuscular characteristics can be used as indicators in predicting reactions and behavior, especially in stressful situations. The nature of this relationship is currently in the realm of speculation, but there is a clear possibility and need for further research on this topic. Also, the results of the correlation analysis showed that a detailed investigation of parameters obtained by analyzing the force-time curve in HGS in relation to psychological characteristics can give new possible supporting insights valuable for the prediction and prevention of maladaptive responses to stress and antisocial behavior among tactical athletes. In this sense, the most illustrative measure is EI. Namely, the high speed and, above all, the short time required for maximum force production, accompanied by a lower possibility of the maximum desired force production, are potentially cofounding factors accompanied by an increased tendency for antisocial behavior among tactical athletes (Figure 1b). Although, overall, MT does not have a large correlation with EI in the general population or tactical athletes (Figure 1a), in female tactical athletes, it is directly related to MT. It is important to note that a fast reaction is often more important in real-life situations encountered by tactical athletes, but that this ability carries with it negative tendencies in behavior and, in the case of women, a weaker motivational component. Another interesting outcome of the analysis is the specificity of tactical athletes, in which increased BMI is accompanied by an increase in MT, while in the general population, it is accompanied by a decrease (Figure 2a). It can be assumed that this is a consequence of the selection of tactical athletes in whom the BMI moves exclusively within the desired limits, and its increase is primarily caused by an increase in muscle mass. When it comes to DT and BMI, there is no specificity between tactical athletes and the general population (Figure 2b).

The obtained findings also suggest that in tactical athletes of both genders, neuromuscular characteristics related to the time and speed of force production are better predictors of psychological characteristics than Fmax. This is not the case with the general population. This finding also contributes to the fulfillment of the practical goals of this research, that is, their application in the selection and training of tactical athletes.

The question of the interpretation intensity of the obtained correlations in this research in relation to previous similar research on this topic arises. When it comes to BMI, the highest correlations reported so far have been with *Conscientiousness* (r = −0.12) [19], while when it comes to HGS, the highest correlations reported so far have been with *Neuroticism* (r = −0.32) [8,9]. In both domains, especially for tactical athletes in our study, higher correlation coefficients with a greater effect were obtained (Table 3 and Table 4). Also, it should be noted that in the field of individual differences, only 25% of studies obtained correlation coefficients greater than 0.3 [39]. These facts support the fact that the innovations we introduced in the research of this topic were justified and that it is justified to use this approach in future research. Nevertheless, one should be careful not to overestimate the obtained results. Although there is a tendency for an association, obtained correlation does not indicate a causal relationship (it does not mean that one variable causes the other). Only 0.6 to 24% of the variation in psychological variables can be explained by handgrip neuromuscular and morphological characteristics. On the other hand, these variables could be part of a wider regression model that could explain a greater percentage of the variance of psychological characteristics, which is a topic for future research.

The findings also provide an opportunity for a theoretical reinterpretation of the nature of the observed associations. Although RFDrel, tFmax, and EI are all complex neuromuscular characteristics, because of their dependence on the speed of the recruitment of motor units, they primarily reflect the ability of the nervous system to quickly produce and transport voluntary impulses [11,12]. The possibility that the characteristics of the nervous system responsible for the speed of impulse generation and transport can, to some extent, be related to or even underlie psychological characteristic development is worthy of additional studies.

This study has certain limitations that should be taken into account when making final conclusions. A convenience sampling method instead of a random one represents a limitation of its representativeness when it comes to the generalization of findings. The same applies to the sample size, especially the male general population group. The deliberate selection of a specific cohort of police and national security students in this study also introduces limitations related to the generalizability of findings. The findings may not be readily applicable to other tactical athlete cohorts, security, or broader populations due to the focused nature of the sample.

The participants, being police and national academy students, might have been inclined to offer responses that aligned with social and academic norms, potentially introducing a skew in the results. Also, correlational design limits the possibility of drawing conclusions about the nature of the observed relationships. For this purpose, new experimental longitudinal studies with more representative samples are required. Lastly, the force measuring device used is not the gold standard, so it would be necessary to check whether the results obtained with its application can be replicated.

## 5. Conclusions

This study was conducted with the aim of exploring the interplay of handgrip neuromuscular, morphological, and psychological characteristics in tactical athletes and the general population of both genders. Numerous significant correlations were obtained, as well as differences between tactical athletes and the general population of both genders. The most prominent obtained correlations were between the *excitation index* with *Psychopathy* and the Dark Triad (ρ = −0.41, −0.39) in female tactical athletes, as well as Narcissism with body height, maximal force, and the maximum rate of force development (ρ = 0.49, 0.43, 0.41) in the male general population. The biggest surprise was the positive correlation of maximal force with both Mental Toughness and Narcissism in the male general population. Also, occupation- and gender-specific patterns were revealed. Although the results of this study indicated the possibility of the existence of correlations between handgrip neuromuscular, morphological, and psychological characteristics in tactical athletes of both genders, nevertheless, at the moment, there is not enough solid evidence for that. That is why new research is needed. The analysis of muscle contractile and time parameters as neuromuscular indicators in the handgrip strength task, which is a novelty compared to previous similar research, proved to be a promising method for future research on this topic. As far as practical application is concerned, the addition of *Mental Toughness* and the *Dark Triad* in selecting future police officers and national security workers is suggested on the basis of the obtained results. These study results could also fit into a broader predictive model of psychological characteristics, which research has yet to produce and which would have significant implications for the selection and training process of tactical athletes.

## Figures and Tables

**Figure 1 jfmk-10-00022-f001:**
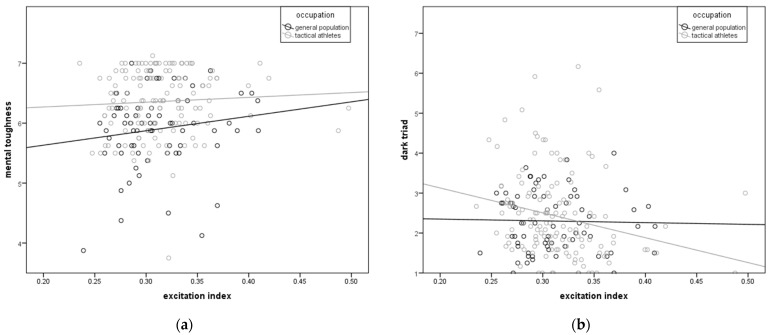
Occupation-based specific patterns of the relationship of handgrip neuromuscular and psychological characteristics among tactical athletes and the general population: (**a**) excitation index and mental toughness and (**b**) excitation index and dark triad.

**Figure 2 jfmk-10-00022-f002:**
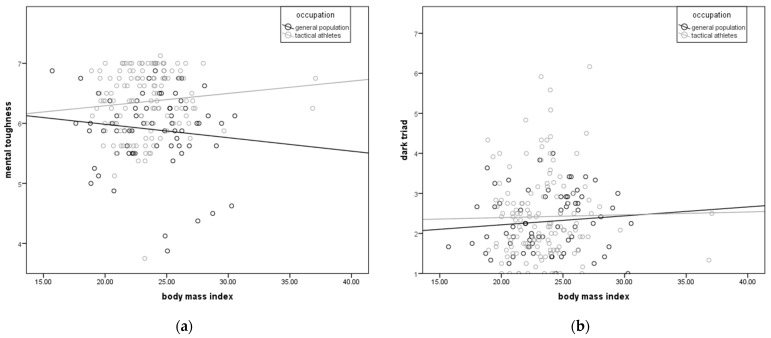
Occupation-based specific patterns of the relationship of morphological and psychological characteristics among tactical athletes and the general population: (**a**) body mass index and mental toughness and (**b**) body mass index and dark triad.

**Table 1 jfmk-10-00022-t001:** Descriptive statistical analysis for the handgrip neuromuscular, morphological, and psychological characteristics of the female subsample.

		MT	Mch	Psc	Nrc	DT	BH	BW	BMI	F_rel_	RFD_rel_	tF_max_	tRFD_max_	DIF	DIRFD	EI
Tactical Athletes (n = 54)	M	6.21	1.75	1.86	3.07	2.21	1.71	65.05	22.3	37.53	235.74	1.88	0.25	1.1	1.05	0.32
	SD	0.6	0.99	0.96	1.41	0.85	0.06	8.68	2.89	4.76	35.22	0.67	0.04	0.16	0.16	0.05
	CV	0.1	0.56	0.52	0.46	0.38	0.03	0.13	0.13	0.13	0.15	0.36	0.17	0.15	0.16	0.15
	Min	3.75	1	1	1	1	1.62	51	18.96	26.46	163.45	0.75	0.19	0.76	0.65	0.24
	Max	7	4.75	4.5	6	4.33	1.85	104	36.85	47.49	304.5	4.75	0.43	1.8	1.5	0.5
General Population (n = 39)	M	5.82	2.01	1.86	2.88	2.16	1.66	63.14	22.8	35.32	228.32	1.64	0.25	1.09	1.1	0.32
	SD	0.7	0.74	0.76	1.4	0.7	0.07	11.12	3.31	5.86	48.52	0.56	0.03	0.12	0.18	0.04
	CV	0.12	0.37	0.41	0.49	0.32	0.04	0.18	0.15	0.17	0.21	0.34	0.12	0.11	0.16	0.13
	Min	4.13	1	1	1	1	1.51	42.5	15.71	26.6	149.31	0.72	0.21	0.91	0.7	0.26
	Max	6.88	3.64	3.67	6	3.64	1.77	87.2	30.53	55.05	367.08	2.89	0.32	1.43	1.59	0.41

Note: MT—mental toughness, Mch—Machiavellianism, Psc—psychopathy, Nrc—narcissism, DT—dark triad, BH—body height, BW—body weight, BMI—body mass index, F_rel_—the relative value of maximal force RFD_rel_—the relative value of maximal rate of force development, tF_max_—time required to reach the maximal force, tRFD_max_—the time required to reach the maximal rate of force development, DIF—maximal force dimorphism index, DIRFD—Dimorphism index of maximal rate of force development, EI—Excitation index, M—mean, SD—standard deviation, CV—coefficient of variation, Min—minimum, Max—maximum.

**Table 2 jfmk-10-00022-t002:** Descriptive statistical analysis for the handgrip neuromuscular, morphological, and psychological characteristics of the male subsample.

		MT	Mch	Psc	Nrc	DT	BH	BW	BMI	F_rel_	RFD_rel_	tF_max_	tRFD_max_	DIF	DIRFD	EI
Tactical Athletes (n = 82)	M	6.46	2.18	2.4	3.18	2.55	1.81	78.91	24	53.92	354.42	1.98	0.24	1.05	1.02	0.31
	SD	0.49	1.31	1.32	1.55	1.17	0.06	10.46	2.41	7.98	55.53	0.9	0.04	0.11	0.15	0.03
	CV	0.08	0.6	0.55	0.49	0.46	0.03	0.13	0.1	0.15	0.16	0.45	0.16	0.1	0.15	0.11
	Min	5.38	1	1	1	1	1.65	54.6	18.89	35.8	230.17	0.62	0.2	0.75	0.73	0.25
	Max	7.13	6.17	7	6.25	6.17	1.94	136.7	37.08	73.02	520.02	5.66	0.46	1.31	1.56	0.42
General Population (n = 30)	M	5.94	2.15	2.18	3.34	2.43	1.83	84.11	25	53.25	357.55	1.54	0.24	1.1	1.14	0.31
	SD	0.61	0.92	0.93	1.37	0.79	0.06	10.79	2.75	10.08	83.22	0.66	0.03	0.13	0.19	0.04
	CV	0.1	0.43	0.42	0.41	0.33	0.03	0.13	0.11	0.19	0.23	0.43	0.14	0.12	0.16	0.13
	Min	3.88	1	1	1	1	1.66	51.6	18.73	31.27	171.46	0.58	0.19	0.88	0.81	0.24
	Max	7	4	4.25	5.75	4	1.97	103	30.27	72.28	458.33	2.97	0.36	1.42	1.55	0.41

Note: MT—mental toughness, Mch—Machiavellianism, Psc—psychopathy, Nrc—narcissism, DT—dark triad, BH—body height, BW—body weight, BMI—body mass index, F_rel_—the relative force, RFD_rel_—maximal rate of force development, tF_max_—time required to reach the maximal force, tRFD_max_—the time required to reach the maximal rate of force development, DIF—maximal force dimorphism index, DIRFD—Dimorphism index of maximal rate of force development, EI—Excitation index, M—mean, SD—standard deviation, CV—coefficient of variation, Min—minimum, Max—maximum.

**Table 3 jfmk-10-00022-t003:** Correlation analysis of handgrip neuromuscular and morphological with psychological characteristics for the female subsample.

		BH	BW	BMI	F_rel_	RFD_rel_	tF_max_	tRFD_max_	DIF	DIRFD	EI
Tactical Athletes (n = 54)	MT	0.11	0.02	−0.07	−0.13	−0.10	0.12	0.02	0.27	0.23	0.05
	Mch	0.06	0.18	0.19	−0.14	0.03	−0.22	−0.12	0.02	−0.09	−0.17
	Psc	−0.03	0.07	0.12	−0.09	0.33 *	−0.02	−0.21	0.15	0.04	−0.41 **
	Nrc	0.16	0.05	0.04	0.13	0.30 *	−0.09	−0.21	−0.14	−0.24	−0.19
	DT	0.11	0.10	0.12	0.01	0.35 **	−0.16	−0.26	−0.04	−0.20	−0.39 **
General Population (n = 39)	MT	−0.07	−0.05	−0.01	0.09	−0.14	0.12	0.26	−0.17	−0.16	0.28
	Mch	0.09	0.11	0.09	−0.28	−0.23	−0.18	0.01	0.04	0.23	0.01
	Psc	0.05	0.01	−0.02	−0.18	−0.10	−0.17	−0.06	0.24	0.27	−0.01
	Nrc	0.33 *	0.12	−0.03	0.10	0.06	−0.21	0.09	0.02	0.17	0.02
	DT	0.17	0.08	0.01	−0.01	−0.01	−0.14	0.02	0.10	0.22	0.00

Note: BH—body height, BW—body weight, BMI—body mass index, F_rel_—the relative value of maximal force, RFD_rel_—the relative value of maximal rate of force development, tF_max_—time required to reach the maximal force, tRFD_max_—the time required to reach the maximal rate of force development, DIF—maximal force dimorphism index, DIRFD—Dimorphism index of maximal rate of force development, EI—Excitation index, MT—mental toughness, Mch—Machiavellianism, Psc—psychopathy, Nrc—narcissism, DT—dark triad, *—Correlation is significant at the 0.05 level (2-tailed), **—Correlation is significant at the 0.01 level (2-tailed).

**Table 4 jfmk-10-00022-t004:** Correlation analysis of handgrip neuromuscular and morphological with psychological characteristics for the male subsample.

		BH	BW	BMI	F_rel_	RFD_rel_	tF_max_	tRFD_max_	DIF	DIRFD	EI
Tactical Athletes (n = 82)	MT	0.22 *	0.18	0.11	0.02	−0.11	0.10	0.21 *	−0.01	−0.18	0.21
	Mch	−0.06	−0.11	−0.08	−0.10	0.09	0.08	−0.35 **	0.13	0.26 *	−0.25 *
	Psc	−0.10	0.01	0.08	−0.03	0.10	−0.05	−0.12	0.01	−0.03	−0.14
	Nrc	−0.01	−0.10	−0.13	−0.08	0.02	−0.09	−0.18	0.07	0.19	−0.18
	DT	−0.07	−0.11	−0.08	−0.06	0.13	−0.02	−0.28 *	0.05	0.15	−0.27 *
General Population (n = 30)	MT	−0.05	−0.07	−0.02	0.37 *	0.33	0.06	−0.04	0.16	0.21	−0.13
	Mch	0.10	−0.08	−0.13	0.35	0.23	0.17	0.05	−0.34	−0.15	0.13
	Psc	0.26	0.25	0.20	0.03	0.10	0.04	−0.03	−0.05	0.12	−0.04
	Nrc	0.49 **	0.27	0.05	0.43 *	0.41 *	0.10	−0.05	−0.15	−0.01	−0.07
	DT	0.42 *	0.30	0.14	0.31	0.28	0.12	0.03	−0.15	0.00	0.01

Note: BH—body height, BW—body weight, BMI—body mass index, F_rel_—the relative value of the maximal force, RFD_rel_—the relative value of maximal rate of force development, tF_max_—time required to reach the maximal force, tRFD_max_—the time required to reach the maximal rate of force development, DIF—maximal force dimorphism index, DIRFD—Dimorphism index of maximal rate of force development, EI—Excitation index, MT—mental toughness, Mch—Machiavellianism. Psc—psychopathy. Nrc—narcissism, DT—dark triad, *—Correlation is significant at the 0.05 level (2-tailed), **—Correlation is significant at the 0.01 level (2-tailed).

## Data Availability

Due to the sensitive nature for national security, only some data are available on request.

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
