# Peer review of "Exploring the Interplay of Handgrip Neuromuscular, Morphological, and Psychological Characteristics in Tactical Athletes and General Population: Gender- and Occupation-Based Specific Patterns"

_jfmk, 2025, doi:10.3390/jfmk10010022_

Round 1

Reviewer 1 Report

Comments and Suggestions for Authors

Thank you for the opportunity to evaluate this scientific article. In an effort to synthesize, my suggestion for the authors concerns the following aspects:

I believe that the title is not correctly formulated. The word civil has slightly different meanings. Some conceptual delimitation would concern the following characteristics: it refers to the citizens of a state or to their legal relations with each other (except for the military and representatives of the Church), as well as to their relations with state bodies and organizations. Since this is not the case in this work, I believe that the word -non-athletes_ would be more appropriate.

Also, also in the title...sex and occupation, it would be more correct to say gender and occupation. Also, a replacement of the mentioned word would be appropriate in the text.

In chapter 2, starting with line 110, the age of the participants is not indicated and of course neither the mean nor the SD. Without explicit mention of the age of the subjects, the research results are not relevant to age influencing physiological, cognitive and behavioral characteristics, determining the validity of the conclusions for the studied group. It also allows for comparison of the results with other similar studies and ensures clarity of the limits of applicability of the conclusions.

Author Response

Comment 1: Thank you for the opportunity to evaluate this scientific article. In an effort to synthesize, my suggestion for the authors concerns the following aspects:

Response 1: Many thanks for your time and effort. Your comments helped us to significantly improve the quality of our work and for that we are especially grateful. Also, many thanks for the up-to-dateness and speed of your review; it means a lot to us.

Comment 2: I believe that the title is not correctly formulated. The word civil has slightly different meanings. Some conceptual delimitation would concern the following characteristics: it refers to the citizens of a state or to their legal relations with each other (except for the military and representatives of the Church), as well as to their relations with state bodies and organizations. Since this is not the case in this work, I believe that the word -non-athletes_ would be more appropriate.

Response 2: We agree. Since some of our subjects from the group that was called civilians, are ex, recreational and even active athletes, instead of the proposal to name this group “non-athletes”, we changed their name to “general population”, both in the title and in the rest of the text.

Comment 3: Also, also in the title...sex and occupation, it would be more correct to say gender and occupation. Also, a replacement of the mentioned word would be appropriate in the text.

Response 3: We agree. We accept this and change it both in the title and in the text.

Comment 4: In chapter 2, starting with line 110, the age of the participants is not indicated and of course neither the mean nor the SD. Without explicit mention of the age of the subjects, the research results are not relevant to age influencing physiological, cognitive and behavioral characteristics, determining the validity of the conclusions for the studied group. It also allows for comparison of the results with other similar studies and ensures clarity of the limits of applicability of the conclusions.

Response 4: We agree. We add asked age data in participants section.

Again thank you very much for your support. Kind regards.

Reviewer 2 Report

Comments and Suggestions for Authors

According to the authors:
"This study aimed to explore the associations between handgrip neuromuscular, morphological and psychological characteristics among tactical athletes and civilians. Obtained data showed the existence of differences between the investigated groups in all three reached domains of characteristics, corroborating the first hypothesis." Furthermore, they state:
"...numerous significant associations, supporting the second hypothesis, were also documented. Ultimately, obtained correlations differ in degree and even direction in subsamples of different sexes and professional orientations, affirmed the validity of the third hypothesis."

This reviewer focuses primarily on the thematic approach, the research problem, and whether the methods are robust enough to address the research question. Finally, the writing and formatting are analyzed. In this context:
#1 At least 12 out of 51 articles are self-citations;
#2 The tables are not self-explanatory, filled with abbreviations without definitions;
#3 The English writing needs a thorough revision to allow for the logical reasoning in the introduction, for instance, as well as the rest of the document, to be properly analyzed;
#4 The presented correlations, although significant, are extremely low (none were strong as mentioned, since the highest was 0.49...);
#5 This study speculates and overemphasizes the existence of correlations between handgrip neuromuscular, morphological, and psychological characteristics in tactical athletes of both sexes, which is not confirmed.

In this sense, I am somewhat disappointed to have accepted the review of this article, which lacks rigor in its methodology, format, and writing.

Comments on the Quality of English Language

The English writing needs a thorough revision to allow for the logical reasoning in the introduction, for instance, as well as the rest of the document, to be properly analyzed;

Author Response

Comment 1: According to the authors:
"This study aimed to explore the associations between handgrip neuromuscular, morphological and psychological characteristics among tactical athletes and civilians. Obtained data showed the existence of differences between the investigated groups in all three reached domains of characteristics, corroborating the first hypothesis." Furthermore, they state:
"...numerous significant associations, supporting the second hypothesis, were also documented. Ultimately, obtained correlations differ in degree and even direction in subsamples of different sexes and professional orientations, affirmed the validity of the third hypothesis."

This reviewer focuses primarily on the thematic approach, the research problem, and whether the methods are robust enough to address the research question. Finally, the writing and formatting are analyzed. In this context:

Response 1: Many thanks for your time and quick response. We appreciate your comments and consider them important in order to improve our work

Comment 2: #1 At least 12 out of 51 articles are self-citations;

Response 2: Thank you for noticing this, although it was not our intention to insert inappropriate self-quotes, this number has been reduced to 3 necessary quotes.

Comment 3: #2 The tables are not self-explanatory, filled with abbreviations without definitions;

Response 3: We add definitions as well as units under every table.

Comment 4: #3 The English writing needs a thorough revision to allow for the logical reasoning in the introduction, for instance, as well as the rest of the document, to be properly analyzed;

Response 4:  We agree, so we used help of Native English speaker PhD from University of California Irvine to improve language.  

Comment 5: #4 The presented correlations, although significant, are extremely low (none were strong as mentioned, since the highest was 0.49...);

Response 5: In method section, subchapter 2.6 Statistical analysis  We stated: “The effect size of correlation coefficients was defined as recommended in guidelines for individual differences studies as weak = 0.10–0.19, moderate = 0.20–0.29, or strong ≥ 0.30 [29].” In the selected study (Gignac, & Szodoraistates, 2016) is stated:  “Individual differences researchers very commonly report Pearson correlations between their variables of interest. Cohen (1988) provided guidelines for the purposes of interpreting the magnitude of a correlation, as well as estimating power. Specifically, r = 0.10, r = 0.30, and r = 0.50 were recommended to be considered small, medium, and large in magnitude, respectively. However, Cohen's effect size guidelines were based principally upon an essentially qualitative impression, rather than a systematic, quantitative analysis of data.” “Based on 708 meta-analytically derived correlations, the 25th, 50th, and 75th percentiles corresponded to correlations of 0.11, 0.19, and 0.29, respectively. Based on the results, it is suggested that Cohen's correlation guidelines are too exigent, as <3% of correlations in the literature were found to be as large as r = 0.50. Consequently, in the absence of any other information, individual differences researchers are recommended to consider correlations of 0.10, 0.20, and 0.30 as relatively small, typical, and relatively large, in the context of a power analysis, as well as the interpretation of statistical results from a normative perspective. “ This approach is adopted in more than 1000 studies in the field in last 10 years.

Never the less, since it is major complaint, we adopted Cohen approach and change interpretation of the correlation in all text. All significant correlation are interpreted as moderate according to Cohen rules. Also we add insights of Gignac, & Szodoraistates study in to the discussion in order to make final conclusion about obtained results as objective as we can.   

Comment 6:#5 This study speculates and overemphasizes the existence of correlations between handgrip neuromuscular, morphological, and psychological characteristics in tactical athletes of both sexes, which is not confirmed.

Response 6: As mentioned in previous comment, we have significantly softened the interpretation of the strength of the correlations, so that instead of strong ones we are talking about moderate correlations. However, statistically significant correlations were obtained, which can be seen in the tables. We also add in the discussion comparison of our findings to prior similar studies. In this regard to all of this, conclusions were drawn.

Comment 7: In this sense, I am somewhat disappointed to have accepted the review of this article, which lacks rigor in its methodology, format, and writing.

We are sorry that you feel this way. We tried to fix all three and we hope you will feel better after the revised version. Certainly, many thanks for your time and effort.

Comment 8: The English writing needs a thorough revision to allow for the logical reasoning in the introduction, for instance, as well as the rest of the document, to be properly analyzed;

Response 8: We agree, so we used help of Native English speaker PhD from University of California, Irvine to improve language.  

THANK YOU SO MUCH AND KIND REGARDS

Reviewer 3 Report

Comments and Suggestions for Authors

The article “Exploring the Interplay of Handgrip Neuromuscular, Morphological, and Psychological Characteristics in Tactical Athletes and Civilians: Sex and Occupation Based Specific Patterns” is interesting and valuable research for the sport science. The article aims to investigate the interplay between neuromuscular, morphological, and psychological characteristics in tactical athletes and civilians, presenting the differentiation of correlations across sex and occupation-based contexts. Despite its valuable objectives, this study has significant limitations and methodological issues that require improvement:

1.              The abstract has several paragraphs, but they are not fully presented, e.g. “Background/Objectives” presents only the objectives, but nothing about study background. In abstract authors use abbreviations, but there are not explained (e.g. HGS, MT, DT0.).

2.              The stated objectives are ambitious, exploring multiple characteristics in both tactical athletes and civilians. The study uses a convenience sample of only 205 participants, leading to concerns about the representativeness and generalizability of the findings. It is not explained how civilians were selected, what study programmes they are representing, do they have anything in common with sports (e.g. Sports and Physical Education study program). Moreover, there is missing information about sample age. Anthropometric data of both groups I would recommend presenting in Material and Methods part. Now the links to the tables presented, but it appears just in Result part, after two pages.

3.              The force measurement methodology lacks robustness. While the authors claim the use of a custom-made device and software for measuring handgrip strength (HGS), validation data provided (ICC values) are insufficient without a thorough discussion of potential biases or limitations in the device's precision.

4.              The statistical analyses are described inadequately. While some statistical methods are mentioned, such as the Mann-Whitney U test and Spearman's correlation, the justification for using these specific tests is weak. The article should include a rationale for these choices, particularly considering the mixed nature of data (categorical vs. continuous) and the non-normality of some variables. Moreover, effect sizes, while mentioned, are not comprehensively discussed. Effect sizes provide essential context to understand the practical significance of differences, particularly in applied research like this one. More attention should be given to effect sizes alongside p-values to ensure readers can assess the real-world relevance of the findings.

5.              2.6. part Title presented differently from others.

6.              Tables and data visualization are insufficient and poorly formatted. Data are presented in a tabular form without providing insights into their interpretation. Below each table is missing notes in which necessary explanations of used abbreviations. Moreover, all the tables seem overloaded with data and abbreviations but there is missing comparison or effects calculations (e.g. 1 table). Additionally, some data used to repeat from the tables in text above the table (e.g. lines 232-242). The use of graphical representation could help illustrate key findings, particularly in exploring the force-time parameters and their psychological correlates.

7.              Links to some tables presented below the table, but not above (e.g. Table 3).

8.              The discussion section lacks critical engagement with existing literature. Although references are provided, the authors fail to critically compare their findings to prior studies. For example, while correlations between handgrip strength and mental toughness are noted, the implications of these findings are not well articulated within the context of existing research.

9.              The article does not address its limitations adequately.

10.           Reference list for such kind of article is appropriate. Cited 51 references and in some introduction and discussion paragraphs is missing at all (267-273 lines). Additionally, I would like to note that just 28 references were published recently (after 2018 year). Some of the references presented not in accordance with the requirements (e.g. 48, 49).

11.           Some grammatical and stylistic mistakes are notable (e.g. line 32, 218). Spaces between words not always correctly presented (e.g. line 71, 89 etc.).

Comments on the Quality of English Language

The article “Exploring the Interplay of Handgrip Neuromuscular, Morphological, and Psychological Characteristics in Tactical Athletes and Civilians: Sex and Occupation Based Specific Patterns” is interesting and valuable research for the sport science. The article aims to investigate the interplay between neuromuscular, morphological, and psychological characteristics in tactical athletes and civilians, presenting the differentiation of correlations across sex and occupation-based contexts. Despite its valuable objectives, this study has significant limitations and methodological issues that require improvement:

1.              The abstract has several paragraphs, but they are not fully presented, e.g. “Background/Objectives” presents only the objectives, but nothing about study background. In abstract authors use abbreviations, but there are not explained (e.g. HGS, MT, DT0.).

2.              The stated objectives are ambitious, exploring multiple characteristics in both tactical athletes and civilians. The study uses a convenience sample of only 205 participants, leading to concerns about the representativeness and generalizability of the findings. It is not explained how civilians were selected, what study programmes they are representing, do they have anything in common with sports (e.g. Sports and Physical Education study program). Moreover, there is missing information about sample age. Anthropometric data of both groups I would recommend presenting in Material and Methods part. Now the links to the tables presented, but it appears just in Result part, after two pages.

3.              The force measurement methodology lacks robustness. While the authors claim the use of a custom-made device and software for measuring handgrip strength (HGS), validation data provided (ICC values) are insufficient without a thorough discussion of potential biases or limitations in the device's precision.

4.              The statistical analyses are described inadequately. While some statistical methods are mentioned, such as the Mann-Whitney U test and Spearman's correlation, the justification for using these specific tests is weak. The article should include a rationale for these choices, particularly considering the mixed nature of data (categorical vs. continuous) and the non-normality of some variables. Moreover, effect sizes, while mentioned, are not comprehensively discussed. Effect sizes provide essential context to understand the practical significance of differences, particularly in applied research like this one. More attention should be given to effect sizes alongside p-values to ensure readers can assess the real-world relevance of the findings.

5.              2.6. part Title presented differently from others.

6.              Tables and data visualization are insufficient and poorly formatted. Data are presented in a tabular form without providing insights into their interpretation. Below each table is missing notes in which necessary explanations of used abbreviations. Moreover, all the tables seem overloaded with data and abbreviations but there is missing comparison or effects calculations (e.g. 1 table). Additionally, some data used to repeat from the tables in text above the table (e.g. lines 232-242). The use of graphical representation could help illustrate key findings, particularly in exploring the force-time parameters and their psychological correlates.

7.              Links to some tables presented below the table, but not above (e.g. Table 3).

8.              The discussion section lacks critical engagement with existing literature. Although references are provided, the authors fail to critically compare their findings to prior studies. For example, while correlations between handgrip strength and mental toughness are noted, the implications of these findings are not well articulated within the context of existing research.

9.              The article does not address its limitations adequately.

10.           Reference list for such kind of article is appropriate. Cited 51 references and in some introduction and discussion paragraphs is missing at all (267-273 lines). Additionally, I would like to note that just 28 references were published recently (after 2018 year). Some of the references presented not in accordance with the requirements (e.g. 48, 49).

11.           Some grammatical and stylistic mistakes are notable (e.g. line 32, 218). Spaces between words not always correctly presented (e.g. line 71, 89 etc.).

Author Response

Comment: The article “Exploring the Interplay of Handgrip Neuromuscular, Morphological, and Psychological Characteristics in Tactical Athletes and Civilians: Sex and Occupation Based Specific Patterns” is interesting and valuable research for the sport science. The article aims to investigate the interplay between neuromuscular, morphological, and psychological characteristics in tactical athletes and civilians, presenting the differentiation of correlations across sex and occupation-based contexts. Despite its valuable objectives, this study has significant limitations and methodological issues that require improvement:

Response: First of all, I want to express my deep gratitude to you in the name of the whole team for your time and suggestions, which are all well-intentioned and help us to improve our work.

Comment 1.              The abstract has several paragraphs, but they are not fully presented, e.g. “Background/Objectives” presents only the objectives, but nothing about study background. In abstract authors use abbreviations, but there are not explained (e.g. HGS, MT, DT0.).

Response 1: Thanks for noticing this. We add background to our abstract:” Correlation of hand-grip strength (HGS) and morphological characteristics with Big Five personality traits are well documented. What is not known do these relationships also exist in highly trained and selected populations such as tactical athletes and are there specificities in relation to the general population?” also, Abbreviation usage is fixed.

Comment 2.              The stated objectives are ambitious, exploring multiple characteristics in both tactical athletes and civilians. The study uses a convenience sample of only 205 participants, leading to concerns about the representativeness and generalizability of the findings. It is not explained how civilians were selected, what study programmes they are representing, do they have anything in common with sports (e.g. Sports and Physical Education study program). Moreover, there is missing information about sample age. Anthropometric data of both groups I would recommend presenting in Material and Methods part. Now the links to the tables presented, but it appears just in Result part, after two pages.

Response 2: We agree that sample size, leading to concerns about the representativeness and generalizability is one of main limitations of the study, we addressed that issue in the closing parts of the Discussion.  Also we add all asked information in Method section:” The research was conducted on a convenient sample of 205 participants (age 22.91 ±7.41, BH=1.76±0.09, BW=73.02±13.04, BMI=23.47±2.91), among whom 93 females (age 22.25±6.25, BH=1.69±0.06, BW=64.25±9.76, BMI=22.51±3.07) and 112 males (age 23.46 ±8.24, BH=1.82±0.06, BW=80.30±10.75, BMI=24.27±2.52), as well as 136 tactical athletes (age 20.12 ±3.44, BH=1.77±0.08, BW=73.41±11.90, BMI=23.33±2.73) and 69 members of general population (age 28.41 ±9.75, BH=1.73±0.11, BW=72.26±15.11, BMI=23.76±3.25). 12 participants were left handed and 1 ambidexter while all others where right-handed.  Participants were students of University of Criminal Investigation and Police Studies in Belgrade, Academy for National Security in Belgrade, members of the general student population (predominantly from Faculty for Special Education and Rehabilitation, and Faculty of sport and physical education, University of Belgrade), as well as the working population of different professional orientations. Participants from general population were volunteers who responded to an invitation placed by the researchers on social networks. Regarding participation in sport, 25.35% members of the general population group declare themselves as non-athletes, 35.21% as recreational athletes, 5.63 as active athletes and 33.80 as former athletes. The group had 5.87±6.09 years of sport and physical activity experience. The criteria for participation were voluntary registration and the absence of any health problems before and during the study. Any history of arm injury was exclusion criteria.”

Comment 3.              The force measurement methodology lacks robustness. While the authors claim the use of a custom-made device and software for measuring handgrip strength (HGS), validation data provided (ICC values) are insufficient without a thorough discussion of potential biases or limitations in the device's precision.

Response 3: We add clarification for this:The HGS was assessed using a custom-made device (SMS HG system) and software system (Isometrics Lite, ver. 3.1.1, Isometrics SMS All4Gym, Belgrade). This system was shown to be valid and reliable for this type of testing compared to Jamar Handgrip Dynamometer considered a gold standard for handgrip testing (dominant hand ICC = 0.98, non-dominant hand ICC = 0.97) [26]. Although validation study showed that results obtained by two devices cannot be compared directly, ant that the instruments cannot be used interchangeably, it also showed that both devices accurately measure the same neuromuscular characteristic of the muscle groups. That is why this system is in vide use in testing of the athletes as well as in research [2,27].  ” Also researches from our lab and us published numerous other studies using this device, but because of auto citation policy of the journal we provide just those 3 references.

Comment 4.              The statistical analyses are described inadequately. While some statistical methods are mentioned, such as the Mann-Whitney U test and Spearman's correlation, the justification for using these specific tests is weak. The article should include a rationale for these choices, particularly considering the mixed nature of data (categorical vs. continuous) and the non-normality of some variables. Moreover, effect sizes, while mentioned, are not comprehensively discussed. Effect sizes provide essential context to understand the practical significance of differences, particularly in applied research like this one. More attention should be given to effect sizes alongside p-values to ensure readers can assess the real-world relevance of the findings.

Response 4: We add clarification of statistical method choiceThe Kolmogorov–Smirnov test was used to assess the normality of distribution, while Levene’s test was used to assess homogeneity of variance. Having in mind mixed nature of data, violation of normal distribution in some variables as well as homogeneity of variance, basic assumptions for usage of parametric statistics are not meet, so it is necessary to apply non-parametric methods in further analyses.This is further described in Results section : “The nonparametric Kolmogorov–Smirnov test showed significant deviations from the normal distribution for the variables Mch and Psc, while Levene’s test show significant violation of variance homogeneity in  Mch, Fmax and RFD.Also we add clarification of effect size and references for it:  “The criterion for evaluation of the effect size in Mann-Whitney U test was: 0.01<η2<0.06 - small, 0.06<η2<0.14 - medium, η2>0.14 - large effect [29].  The effect size of correlation coefficients was defined as recommended in guidelines for individual differences studies as weak = 0.10–0.29, moderate = 0.30–0.49, or strong ≥ 0.50 [30,31]. “  We present calculations of effect size its evaluation in Results. “The non-parametric Mann-Whitney U test reveals significant differences with medium effect size between the female tactical athletes and general population in, MT (U=742, z=-2.4, p=0.01, η2=0.06), Mch (U=737.5, z=-2.5, p=0.01, η2=0.07), BH (U=651, z=-3.1, p=0.02, η2=0.10), Fmax (U=749.5, z=-2.4, p=0.02 η2=0.06)....”

In male sample the non-parametric Mann-Whitney U test reveals significant differences with large effect size in MT (U=595.5, z=-4.2, p<0.01, η2=0.15), medium effect size in BW (U=762, z=-3.1, p=0.02, η2=0.08), tF (U=847, z=-2.5, p=0.01, η2=0.06), and DIRFD (U=785, z=-2.9, p<0.01, η2=0.07) as well as small effect size in BMI (U=859, z=-2.4, p=0.01 η2=0.05), DIF (U=917, z=-2, p=0.04, η2=0.03)..”

“When it comes to gender differences in psychological characteristics, the non-parametric Mann-Whitney U test didn’t revile significant differences between the male and female general population, but in the same time reviled significant differences with small effect size in police-security students in MT (U=1655.5, z=-2.5, p=0.01, η2=0.04) and Psc (U=1649.5, z=-2.5, p=0.01, η2=0.04)....”  

“The correlation analysis in female subsample (Table 3) revealed a moderate effect association of MT with DIF which was on the edge of statistical significance (p=0.05), and statistically significant moderate effect association of Psc, Nrc, and DT with RFD, as well as Psc, and DT with EI, among police and security students. When it comes to the general population, moderate correlation of Nrc with BH were also reviled...”

“The results of correlation analysis in male subsample (Table 4) among tactical athletes revealed moderate effect associations of MT with BH, tRFD, EI which were on the edge of statistical significance (p=0.05), as well as statistically significant moderate effect associations of Mch with DIRFD and EI, DT with tRFD and EI, as well as Mch and RFD. When it comes to the general population, moderate effect correlation of MT with Fmax, Nrc with BH, Fmax and RFD, as well as DT with BH were also reviled. A number of other moderate associations were found in this group but did not cross the threshold of statistical significance (due to small sample size).”

Additionaly, in the discussion we also explained effect sizes of obtained results and compare it to prior similar studies.

Comment 5.              2.6. part Title presented differently from others.

Response 5: We fixed formatting, thanks for noticing.

Comment 6.              Tables and data visualization are insufficient and poorly formatted. Data are presented in a tabular form without providing insights into their interpretation. Below each table is missing notes in which necessary explanations of used abbreviations. Moreover, all the tables seem overloaded with data and abbreviations but there is missing comparison or effects calculations (e.g. 1 table). Additionally, some data used to repeat from the tables in text above the table (e.g. lines 232-242). The use of graphical representation could help illustrate key findings, particularly in exploring the force-time parameters and their psychological correlates.

Response 6: Again agree. We add aberrations explanation as well as units under every table. We add key insights about data interpretation below each table, as well as significant comparison and effect size. We also add two graphs to illustrate Occupation based specific patterns of  relationship of handgrip neuromuscular,  morphological and psychological characteristics among tactical-athletes and general population.

Comment 7.              Links to some tables presented below the table, but not above (e.g. Table 3).

Response 7: We fix that.

Comment 8.              The discussion section lacks critical engagement with existing literature. Although references are provided, the authors fail to critically compare their findings to prior studies. For example, while correlations between handgrip strength and mental toughness are noted, the implications of these findings are not well articulated within the context of existing research.

Response 8: We addressed all named issues in discussion:

We add explanation of obtained effects: “When compared to results of previous studies [18,22] obtained profile can be described as benevolent and stress resilient. Although our research clearly documented the difference between the subsamples of tactical athletes and general population, in favor of the greater development of the first group in all domains, it seems that the general population group is positioned in all three domains above what was expected based on previous research. The only consistent difference with large effect was obtained in MT. This finding could be interpreted in two ways, as a consequence of the selection of tactical athletes, but also as a consequence of the influence of training on development of MT. Most likely, both explanations are valid and represent an opportunity for further research on this topic. Never the less, results of descriptive (Tables 1 and 2) and Mann Whitney u test, suggests that the general population group, since they applied themselves for the research, which was conducted at the Faculty of Sports, actually has an above average experience in sport and physical exercise.”

We further explicated obtained results: “Mentioned finding supports the earlier conclusion that the necessary expansion of the domain of assessment of personality characteristics with B5 is the addition of MT and DT if one wants to understand the predicted reaction to chronic stress in a valid way [38]. This finding should be taken seriously by practitioners working on the selection of tactical athletes, because some superior neuromuscular functioning is related to negative behavioral tendencies, both in man and woman. The results of the correlation analysis showed that a detailed investigation of parameters obtained with analyzing force-time curve in HGS in relation to the psychological characteristics can give new insights valuable in predicting and prevention maladaptive response to stress and antisocial behavior among tactical athletes. In this sense, the most illustrative is EI. Namely, the high speed and, above all, the short time required for maximum force production, accompanied by a lower possibility of maximum desired force production, is accompanied by an increased tendency for antisocial behavior among tactical athletes (Figure 1b). Although overall, MT does not have a large correlation with EI in general population and tactical athletes (Figure 1a), in tactical athletes woman it is directly related with MT. It is important to note that quick reaction is often more important in real situations in which tactical athletes work, but that this ability carries with it negative tendencies in behavior and, in the case of women, a weaker motivational component. Another interesting outcome of the analysis is also the specificity of tactical athletes, in which increased BMI is accompanied by an increase in MT, while in the general population it is accompanied by a decrease (Figure 2a). It can be assumed that this is a consequence of the selection of tactical athletes in whom the BMI moves exclusively within the desired limits, and its increase is primarily caused by an increase in muscle mass. When it comes to DT and BMI, there is no specificity between tactical athletes of the general population (Figure 2b).”

Obtained results and compare it to prior similar studies. “The question arises of the size of the obtained correlations in this research in relation to previous similar researches on this topic. When it comes to BMI, the highest correlations reported so far have been with Conscientiousness (r=-0.12) [19], while when it comes to HGS, the highest correlations reported so far have been with Neuroticism (r=-0.32) [8,9]. In both domains, especially for tactical athletes in our study, higher correlation coefficients with a greater effect were obtained (Tables 3 and 4). Also it should be noted that in the field of individual differences only 25% of studies obtained correlation coefficients greater than 0.3 [38]. These facts support the fact that the innovations we introduced in the research of this topic were justified and that it is justified to use this approach in further research.”

Comment 9.              The article does not address its limitations adequately.

Response 9:  We try to fix this as well: “This study has certain limitations which should be taken into account when making final conclusions. A convenience sampling method instead of a random one represents a limitation of its representativeness when it comes to generalization of findings. The same applies to the sample size, especially the male general population group. The deliberate selection of a specific cohort of police and national security students in this study introduces also limitations related to the generalizability of findings. The findings may not be readily applicable to other tactical athlete’s cohorts or broader populations due to the focused nature of the sample. This restricts the ability to make sweeping conclusions about attitudes and behaviors in the wider context, particularly when influenced by socio-demographic factors. Furthermore, the use of psychological surveys opens the door to response bias, as participants may not always provide entirely accurate or candid responses. The participants, being police and national academy students, might have been inclined to offer responses that aligned with social and academic norms, potentially introducing a skew in the results. Also, correlational design limits the possibility to draw conclusions about the nature of observed relationship. For this purpose, new experimental longitudinal studies with more representative sample are required. Lastly, the force measuring device is not the gold standard, so it would be necessary to check whether the results obtained with its application will be replicated.

Comment 10.           Reference list for such kind of article is appropriate. Cited 51 references and in some introduction and discussion paragraphs is missing at all (267-273 lines). Additionally, I would like to note that just 28 references were published recently (after 2018 year). Some of the references presented not in accordance with the requirements (e.g. 48, 49).

Response 10: We've fixed formatting, reduced the number of auto-citations, and replaced old references with newer ones.

Comment 11.           Some grammatical and stylistic mistakes are notable (e.g. line 32, 218). Spaces between words not always correctly presented (e.g. line 71, 89 etc.).

Response 11: We agree, so we used help of Native English speaker PhD from University of California, Irvine to improve language.  

THANK YOU SO MUCH AND KIND REGARDS

Round 2

Reviewer 1 Report

Comments and Suggestions for Authors

It is ok now

Author Response

Comment 1: It is ok now

Response 2: Thank you very much!!!

Reviewer 2 Report

Comments and Suggestions for Authors

Despite some improvements, the main issue is that the presented correlations, although significant, are extremely low (none were strong as mentioned, since the highest was 0.49...)... In fact, a correlation of at most 0.49 is generally not considered "large"....  For example, 0.40, although there is a tendency for a positive association, a correlation of 0.40 does not indicate a causal relationship (it does not mean that one variable causes the other). Approximately 16% of the variation in one variable can be explained by the other (this is obtained by squaring the coefficient: 0.40= 0.16.... Well, this study speculates and overemphasizes the existence of correlations between handgrip neuromuscular, morphological, and psychological characteristics in tactical athletes of both sexes, which is not confirmed. 

Author Response

Comment 1: Despite some improvements, the main issue is that the presented correlations, although significant, are extremely low (none were strong as mentioned, since the highest was 0.49...). In fact, a correlation of at most 0.49 is generally not considered "large".... 

Response 1: Again, thanx for your time and patients with our text, we agree with all your comments and we will try to fix all that you asked since we believe it truly make our paper better by scientific standards.

We rephrase explanation of all our correlations from moderate to weak.

Comment 2: For example, 0.40, although there is a tendency for a positive association, a correlation of 0.40 does not indicate a causal relationship (it does not mean that one variable causes the other). Approximately 16% of the variation in one variable can be explained by the other (this is obtained by squaring the coefficient: 0.40= 0.16.... 

Response 2: We add this explication in the discussion.  “Never the less, one should be careful not to overestimate the obtained results. Although there is a tendency for an association, obtained correlation does not indicate a causal relationship (it does not mean that one variable causes the other). Only 0.6 to 24% of the variation in psychological variables can be explained by the handgrip neuromuscular and morphological characteristics.”

Comment 3: Well, this study speculates and overemphasizes the existence of correlations between handgrip neuromuscular, morphological, and psychological characteristics in tactical athletes of both sexes, which is not confirmed. 

Response 3: We rephrase our conclusion in order not to overemphasizes the existence of correlations: “Although results of this study indicated possibility of the existence of correlations between handgrip neuromuscular, morphological and psychological characteristics in tactical athletes of both genders, never the less at the moment there is not enough solid evidence for that.”

Reviewer 3 Report

Comments and Suggestions for Authors

Morphological, and Psychological Characteristics in Tactical Athletes and General Population: Gender and Occupation Based Specific Patterns ” has considered most of my remarks made during my first review process. Nevertheless, in my opinion, there are some points to must be fixed additionally:

1.     In my opinion, there is too much data presented in some places in the text (e.g. lines 130-135).

2.     Additionally, it is still very complicated to understand the data in tables (e. g. what type of comparison and significant difference was estimated? Why some data are in bold presented with the sign “*” and some not (e.g. table 4)?

3.     In conclusion, not to use abbreviations (e.g. HGS, MT, DT).

4. Some references presented not in accordance the requirement (e.g. 39).

Author Response

Comment: Morphological, and Psychological Characteristics in Tactical Athletes and General Population: Gender and Occupation Based Specific Patterns ” has considered most of my remarks made during my first review process. Nevertheless, in my opinion, there are some points to must be fixed additionally:

Response: Again, thanx for your time and patients with our text, we agree with all your comments and we will try to fix all that you asked since we believe it truly make our paper better by scientific standards

Comment 1.     In my opinion, there is too much data presented in some places in the text (e.g. lines 130-135).

Response 1: We reduced amount of data present in results.

Comment 2.     Additionally, it is still very complicated to understand the data in tables (e. g. what type of comparison and significant difference was estimated? Why some data are in bold presented with the sign “*” and some not (e.g. table 4)?

Response 2: We try to fix our tables to be easier for reading and understanding. Now bold is removed as being redundant and confusing, also “*” signs have more clear explication: , * - Correlation is significant at the 0.05 level (2-tailed), ** - Correlation is significant at the 0.01 level (2-tailed)

Comment 3.     In conclusion, not to use abbreviations (e.g. HGS, MT, DT).

Response 3: We remove abbreviations in conclusion

Comment 4. Some references presented not in accordance the requirement (e.g. 39).

Response 4: We fix the references manually since Zotero have made some mistakes in referencing

Round 3

Reviewer 2 Report

Comments and Suggestions for Authors

I am happy with the changes proposed by the authors.